# The Antioxidant and Prebiotic Activities of Mixtures Honey/Biomimetic NaDES and Polyphenols Show Differences between Honeysuckle and Raspberry Extracts

**DOI:** 10.3390/antiox12091678

**Published:** 2023-08-28

**Authors:** Luminița Dimitriu, Diana Constantinescu-Aruxandei, Daniel Preda, Ionuț Moraru, Narcisa Elena Băbeanu, Florin Oancea

**Affiliations:** 1Bioproducts Team, Bioresources Department, National Institute for Research & Development in Chemistry and Petrochemistry—ICECHIM, Splaiul Independent, ei No. 202, Sector 6, 060021 Bucharest, Romania; luminita.dimitriu@icechim.ro (L.D.); daniel.preda@stud.chimie.upb.ro (D.P.); 2Faculty of Biotechnologies, University of Agronomic Sciences and Veterinary Medicine of Bucharest, Marasti Blv., No. 59, Sector 1, 011464 Bucharest, Romania; narcisa.babeanu@biotehnologii.usamv.ro; 3Department of Analytical Chemistry and Environmental Engineering, Faculty of Chemical Engineering and Biotechnologies, University Politehnica Bucharest, Str. Gheorghe Polizu nr. 1-7, Sector 1, 011061 Bucharest, Romania; 4Medica Laboratories, Str. Frasinului nr. 11, 075100 Otopeni, Romania; ionut.moraru@pro-natura.ro

**Keywords:** honey, biomimetic natural deep eutectic solvents, *Lonicera caprifolium*, lactic acid production, *Limosilactobacillus reuteri*

## Abstract

In our previous research, we demonstrated that honey and its biomimetic natural deep eutectic solvent (NaDES) modulate the antioxidant activity (AOA) of the raspberry extract (RE). In this study, we evaluated the AOA behaviour of the mixture honey/NaDES–honeysuckle (*Lonicera caprifolium,* LFL) extract and compared it with the mixture honey/NaDES–RE. These two extracts have similar major flavonoids and hydroxycinnamic acid compounds but differ in their total content and the presence of anthocyanins in RE. Therefore, it was of interest to see if the modulation of the LFL polyphenols by honey/NaDES was similar to that of RE. We also evaluated the prebiotic activity of these mixtures and individual components on *Limosilactobacillus reuteri* DSM 20016. Although honey/NaDES modulated the AOA of both extracts, from synergism to antagonism, the modulation was different between the two extracts for some AOA activities. Honey/NaDES mixtures enriched with LFL and RE did not show significant differences in bacterial growth stimulation. However, at a concentration of 45 mg/mL, the honey -LFL mixture exhibited a higher effect compared to the honey–RE mixture. The antioxidant and prebiotic properties of mixtures between honey and polyphenol-rich extracts are determined by multiple interactions in complex chemical systems.

## 1. Introduction

Honey, being a natural product, displays a diverse range of biological activities such as antioxidant, antimicrobial, anti-inflammatory, cytoprotective, prebiotic, and postbiotic [1]. These diverse ranges of biological activities arise from the complex composition of honey, which includes carbohydrates, amino acids, phenolic compounds, minerals, enzymes, and electrolytes [2]. Furthermore, honey exhibits characteristics similar to those of a natural deep eutectic solvent (NaDES) due to the intermolecular interactions between its monosaccharides and disaccharides, as well as the hydrogen bonds formed between them [3,4,5]. Initially, NaDESs were introduced in green chemistry as a viable and eco-friendly alternative to conventional organic solvents. Their distinct properties, including bioavailability, biodegradability, and cost-effectiveness, captured the attention and motivated researchers to assess their potential applicability in the food sector for creating innovative functional food products. Moreover, their relatively more challenging removal after extraction has contributed to their investigation as functional ingredients in the food industry [6,7,8]. Additionally, other characteristics such as water activity, pH, antimicrobial activity, and enzyme interactions play an essential role in the storage and stabilization of food compounds when utilizing NaDES in food applications [9,10]. Honey possesses notable prebiotic properties, making it beneficial for the growth and activity of beneficial gut bacteria [11,12,13]. Prebiotics are non-digestible substances that selectively promote the growth and activity of beneficial microorganisms in the gastrointestinal tract. The prebiotic effects of honey are primarily attributed to its carbohydrate composition, especially oligosaccharides [14].

The antioxidant activity of honey is attributed to its rich content of phenolic acids, flavonoids, and other phenolic compounds. The antioxidant activity of honey helps protect cells from oxidative damage by neutralizing harmful free radicals [15,16,17]. On the other hand, the prebiotic activity of honey refers to its ability to selectively promote the growth and activity of beneficial gut bacteria. There appears to be an interplay between the antioxidant and prebiotic activities of honey. The presence of healthy gut microbiota is essential for the effective absorption and utilization of dietary antioxidants. Beneficial gut bacteria can metabolize certain components of honey, releasing bioactive compounds that contribute to its antioxidant potential. In turn, the antioxidants present in honey can help protect the gut microbiota from oxidative stress, maintaining a balanced microbial community [14,18]. In this case, a good solution would be to improve the biological and especially antioxidant properties of honey.

Plants are a rich source of bioactive compounds, including polyphenols which have various properties (antioxidant, antimicrobial, prebiotic, and others) with different applications in human health and industry [19]. Honeysuckle (*Lonicera caprifolium*) is a perennial flowering plant native to Europe and belongs to the family *Caprifoliaceae*. The honeysuckle flowers have a history of use in traditional herbal medicine due to their antibacterial, antioxidant, and antiviral activities [20]. While several species of the Lonicera genus, such as *L. japonica* (Japanese honeysuckle) and others, have been extensively studied and utilized in traditional medicine and cosmetics, the chemical composition of European honeysuckle (*L. caprifolium*) has received less research attention.

In our recent research [5], we formulated a NaDES that mimics the composition of honey by incorporating essential sugars found in honey (glucose, fructose, and sucrose). The NaDES derived from this formulation was analyzed, comparing its structural and physicochemical properties with honey. Our findings revealed that the honey-biomimetic NaDES closely resembled honey in terms of its characteristic features. Within the same study, we improved the antioxidant potential of honey and its biomimetic DES by incorporating dried raspberry extract and standard polyphenols found in the raspberry extract (caffeic acid and epicatechin) and evaluated the interaction in terms of antioxidant activity between them (between honey/NaDES and polyphenols). The main scope of analysing honey in comparison to NaDES was to understand better the behaviour of honey-polyphenols mixtures and test if the polysaccharides composition and interactions were sufficient to explain this behavior. A better understanding of honey properties could also help to design an edible or biocompatible biomimetic product based on its NaDES characteristics.

In this study, we aimed to explore the antioxidant activity of formulations between honey/biomimetic NaDES and polyphenols extracted from honeysuckle flowers. Building on previous research, we investigated how incorporating dried extract of honeysuckle flowers could enhance the antioxidant activity (AOA) of honey and its biomimetic NaDES. Our focus was on evaluating the AOA of this new mixture and understanding the interactions between its components.

Furthermore, we compared the AOA behaviour of these honey/NaDES mixtures with those enriched with raspberry extract to determine their relative efficacy. Alongside this, we conducted an assessment of the prebiotic activity of these mixtures and their individual components using the strain *Limosilactobacillus reuteri* DSM 20016. To evaluate prebiotic activity, we measured growth activity and determined L-lactic acid production as metabolites during fermentation.

## 2. Materials and Methods

### 2.1. Materials

Fresh honeysuckle flowers (*Lonicera caprifolium*, *family Caprifoliaceae*) were harvested in Bucharest area, identified based on their morphological characteristics, and a herbarium voucher with the number [USAMV B 4093] was deposited in the herbarium of USAMV Bucharest. These flowers and multifloral honey (RomHoney Group, Iasi, Romania) were used in this work. The multifloral honey was prepared by mixing 1/3 rapeseed honey with 1/3 sunflower honey and 1/3 meadow honey. The honeysuckle flowers were dried at room temperature and were ground to a fine powder using an electrical grinder. The following chemicals were used: ethanol 96% (Reactivul București Srl, Bucharest, Romania), D(+)-Glucose anhydrous extra pure, D(−)-Fructose, extra pure, D(+) Saccharose, reagent grade (Scharlau, Barcelona, Spain) Trolox 97% (Acros Organics, Thermo Fisher Scientific, Pittburghs, PA, USA), 2,2-Diphenyl-1-picrylhydrazyl (Sigma-Aldrich, Merck Group, Darmstad, Germany), 2,20-Azino-bis(3-ethylbenzothiazo-line-6-sulfonic acid) diammonium salt, 98%, 2,4,6-tri (2-pyridyl-1,3,5-triazine) 98% (Alfa Aesar, Kandel, Germany), Folin Ciocalteu’s phenol reagent, Iron chloride (III) (Merck, Darmstadt, Germany), hydrochloric acid, acetic acid (Chimopar Srl, Bucharest, Romania), sodium acetate, MRS broth and aga (Scharlau, Barcelona, Spain), HPLC standards: ferulic acid, p-coumaric acid, caffeic acid, quercetin dihydrate (Sigma-Aldrich, Merck Group, Darmstad, Germany), syringic acid, luteolin, (+)-rutin trihydrate, (Alfa Aesar, Haverhill, MA, USA), chlorogenic acid, myricetin (Cayman Chemical, Ann Arbor, MI, USA), apigenin, (−) epicatechin (Roth, Karlsruhe, Germany), and kaempferol (Cayman Chemical, Ann Arbor, MI, USA). K-DLATE kit for D-/L-Lactic Acid (D-/L-Lactate) (Rapid) Assay Kit (Megazyme, Wicklow, Ireland).

### 2.2. Hydroalcoholic Extraction of Polyphenols from Honeysuckle Flower

The polyphenols extraction from honeysuckle flowers (*L. caprifolium*) was performed according to the method described by [21]. The polyphenol compounds were extracted using an ultrasound-assisted method in 61% (*v*/*v*) ethanol, in the ratio 1:20 (plant material/solvent) and 30 min of reaction in an ultrasound bath. The supernatant was removed after centrifugation at 7350 rcf for 30 min, and extraction was repeated in the same condition described below.

### 2.3. Analysis of Polyphenolic Content of Honeysuckle Extract 

#### 2.3.1. Total Polyphenol Content

The total polyphenol content (TPC) of the honeysuckle flower extract was measured spectrophotometrically by the Folin–Ciocalteau assay according to [22]. The method involved mixing 0.01 mL of honeysuckle extract or standard solutions of gallic acid with 0.09 mL double-distilled water (ddH_2_O) and 0.010 mL of Folin–Ciocalteu reagent. After 5 min of reaction, 0.1 mL of 7% Na_2_CO_3_ and 0.04 mL ddH_2_O were added to the mixture and incubated at room temperature for 60 min. The absorbance of solutions was measured at 765 nm using a plate reader (CLARIOstar, BMG LABTECH, Ortenberg, Germany). The calibration curve was performed using different concentrations of gallic acid in 70% (*v*/*v*) of ethanol. The range of gallic acid concentrations used was between 5 and 30 µg/mL. The results of TPC were calculated and reported as mg gallic acid equivalent/100 g dry weight of the sample (mg GAE/100 g DW).

#### 2.3.2. Total Hydroxycinnamic Acid Content

The total hydroxycinnamic acid content (HAT) of honeysuckle flower extract was quantified spectrophotometrically according to the method adapted from the European Pharmacopoeia [23]. The method involved mixing 0.025 mL of honeysuckle extract or standard solutions of chlorogenic acid with 0.05 mL of 0.5 M HCl and with 0.05 mL of a solution composed of 1% (*w*/*v*) NaNO_2_ and 1% (*w*/*v*) Na_2_MoO_4_, followed by 0.05 mL of 8.5% NaOH and 0.07 mL ddH_2_O. The absorbance of solutions was measured at 524 nm. The calibration curve was performed using different concentrations (0–50 µg/mL) of chlorogenic acid in 70% (*v*/*v*) of ethanol. The HAT of the honeysuckle flower was expressed as mg chlorogenic acid equivalent/100 g DW of the sample (ChaE mg/100 g DW). 

#### 2.3.3. Total Flavonoid Content

Determination of the total flavonoid content (TFC) of the honeysuckle flower extract extracts was performed using the aluminium chloride/sodium acetate method according to [24] with some modifications. The method involved mixing 0.1 mL of honeysuckle extract or standard solution of quercetin with 0.1 mL of 10% CH3COONa and 0.12 mL of 2.5% AlCl_3_ as well as 0.68 mL of ddH_2_O were added to the mixture. The absorbance of the mixture was measured at 430 nm after 45 min. The results of TFC were calculated as quercetin equivalent mg/100 g DW of the sample (QE mg/100 g DW).

#### 2.3.4. Total Anthocyanin Content

The total anthocyanin content (TAC) was determined using the pH differential method [25]. In brief, the absorbance of 2.5 × diluted sample in 25 mM potassium chloride buffer at pH 1 and 0.4 M sodium acetate buffer at pH 4.5 was measured after 30 min of incubation at room temperature, at 520 nm and 700 nm using a UV-VIS-NIR spectrophotometer (Ocean Optics, Orlando, FL, USA). The TAC was calculated by the following equation: TAC = (ΔAs × MW × DF × V × 1000)/(ε × L × m), where ΔAs—difference of the absorbance of the sample at pH 1 and pH 4.5, DF is the dilution factor, L—optical pathlength (1 cm), V—the volume of the extracts (L), ε—molar absorptivity coefficient and MW—the molecular weight of cyanidin 3-glucoside (ε = 26,900 M ^−1^ cm^−1^ and MW = 449.2 g/mol), ΔAs = (A_520_ − A_700_)pH1.0 − (_A520_ − A_700_)pH4.5. The result was expressed as milligrams of cyanidin 3-glucoside equivalent per 100 g of dry weight (DW) of the sample (mg cya, 3-Gluequivalent/100 g DW).

#### 2.3.5. HPLC Analysis

The high-pressure liquid chromatography (HPLC) analysis of polyphenolic compounds from honeysuckle flower extract was carried out on Dionex Ultimate 3000 system (Thermo Fisher Scientific, Waltham, MA, USA) equipped with VWD-3100 detector. Data processing and analysis were performed by Chromelleon 7.0 software (Thermo Fisher Scientific, Waltham). 

*HPLC Analysis of Phenolic Acids*. The phenolic acids content and composition of honeysuckle flower extract were determined by HPLC analysis according to a method described by [25] with some modifications. Chromatographic separation was performed on a Luna Omega 5 µm Polar C18 100 Å column (250 mm × 4.6 mm) (Phenomenex, Torrance, CA, USA). The mobile phase consisted of an aqueous solution with 0.1% formic acid (solvent A) and methanol (solvent B). The total runtime of the method was 40 min with the following gradient elution program: 0–25 min. 5% B/95% A, 25–33 min. 30% B/70% A, 34–40 min. 5% B/95% A. The analysis was conducted at a constant flow rate of 1.25 mL × min^−1^, and the injection volume was set to 10 µL. The phenolic acids were detected at λ = 280 nm. 

The identification of the phenolic acids involved comparing them with standards for each identified compound based on the retention time of standards. Quantification was accomplished by creating calibration curves for each determined compound using the standards. These calibration curves exhibited excellent linearity (R^2^ = 0.9996) when plotting peak area against concentration and were in the range of 18.125–1000 µg/mL.

*HPLC Analysis of Flavonoids*. The composition and quantification of flavonoids from the extract of honeysuckle flower were determined by HPLC analysis according to the method described by [26]. The separation of flavonoids was performed on an Omega 5 µm Polar C18 100 Å column (250 mm × 4.6 mm) (Phenomenex, Torrance). The method involved using a gradient elution of two solvents: Methanol (solvent A) and 0.5% H_3_PO_4_ (solvent B). The gradient elution program was set as follows: 0–10 min 15% A/85% B, 15–25 min 85% A/15% B, 25–30 min. 60% A/40% B. The flow rate of the mobile phase was 1.5 mL/min, and the column temperature was 25 °C to detect flavonoids at 280 nm.

Flavonoids were detected and measured by correlating their retention time and spectral properties with established standards through the utilization of a calibration curve.

### 2.4. Preparation of a Mixture of Honey/GFSw with Honeysuckle Extract

The biomimetic NaDES, abbreviated from this point onwards GFSw (glucose/fructose/sucrose/water—the components of NaDES), was prepared by the method described in our previous work [5]. Honey/GFSw mixtures enriched with dried *L. caprifolium* extract were prepared in the same way as the mixtures with a raspberry extract from our previous study [5]. The honeysuckle flower extract was divided into three equal fractions, and each fraction was then concentrated to dryness (E_CD) at 40 °C using a semi-automated evaporation system called MultiVap54 (Lab tech, Sorisole, Italy). Two of the fractions E_CD were resuspended in honey (H) and, respectively, in its biomimetic NaDES named GFSw from this point onwards, at a ratio of 1:20 (*w*/*w*), resulting in the honey-honeysuckle mixture sample (H_LFL) and GFSw_LFL. The last fraction of E_CD was resuspended in 70% ethanol solution at the same ratio as in honey/GFSw (1:20 *w*/*v*), resulting in the LFL sample. The E_CD was dissolved in honey/GFSw, subjecting it to an ultrasonic bath, ensuring thorough mixing, and allowing the polyphenols to diffuse into the honey/GFSw overnight. 

### 2.5. Antioxidant Activity 

For the assessment of antioxidant activity (AOA), the H/GFSw and H_LFL/GFSw_LFL samples were dissolved in 70% (*v*/*v*) ethanol at a concentration of 0.2 g/mL (*w*/*v*). Four spectrophotometric methods, namely radical scavenging activity (ABTS and DPPH) and reducing antioxidant power (CUPRAC and FRAP), were employed to measure the AOA of the samples.

The AOA analysis was conducted at various concentrations of the samples, and calibration curves of the samples were generated for each method. The concentration values of LFL were individually tested and matched with the concentrations of LFL present in mixtures containing honey/GFSw. The final concentrations used in the final testing encompassed a range of 2 to 200 mg/mL for honey or GFSw and their mixtures and 0.1 to 10 mg/mL for LFL.

#### 2.5.1. Radical Scavenging Activity by the DPPH Assay

The radical scavenging activity of the samples was tested by the DPPH method as described by [27] with slight modifications. To 100 µL of the sample, 100 µL of 0.3 mM DPPH solution dissolved in 99.6% (*v*/*v*) ethanol. The samples were incubated in the dark at room temperature for 30 min. The absorbance of the reaction mixture was measured at 517 nm using a UV-Vis plate reader (CLARIOstar, BMG LABTECH, Ortenberg, Germany).

#### 2.5.2. Radical Scavenging Activity by the ABTS Assay

The ABTS cation scavenging activity of the samples was evaluated using the method adopted by [28]. ABTS radical cation solution was produced by mixing 7 mM ABTS in H_2_Od.d. and 2.45 mM potassium persulfate solution. This solution was left for 12–16 h before being used in the dark at room temperature. Before use, the ABTS^+^ solution was diluted with 96% ethanol to obtain an absorbance of 0.700 ± 0.04 at 734 nm. An aliquot of 0.02 mL of the sample was added to 0.180 mL of diluted ABTS^+^ solution, and the absorbance was read at 734 nm after 30 min of incubation in the dark at room temperature.

#### 2.5.3. Ferric-Ion Reducing Antioxidant Power (FRAP) Assay

The FRAP assay was performed according to the procedure described by [29] with slight modifications. The FRAP reagent was composed of 0.3 M acetate buffer at pH 3.6, 0.01 M TPTZ (solubilized in 0.04 mM HCl), and 0.02 M FeCl_3_ solution in the ratio 10:1:1, which was warmed at 37 °C before to use. The method involved mixing 15 µL of sample/standard solution of Trolox with 285 µL freshly prepared FRAP reagent. The reaction mixtures were incubated at 37 °C in the dark for 30 min, and the absorbance was measured at 593 nm. The calibration curve was developed using different concentrations (50–450 µg/mL) of Trolox in 70% (*v*/*v*) of ethanol. 

#### 2.5.4. Cupric-Ion Reducing Antioxidant Capacity (CUPRAC) Assay

The CUPRAC method was performed according to the adopted procedure described by [30]. An aliquot of 10 µL of the samples/standard solutions of Trolox was mixed with 30 µL CuSO_4_ (5 mM), 30 µL neocuproine (3.75 mM), and 280 µL distilled H_2_Od.d. The absorbance was measured at 450 nm after 30 min of incubation at room temperature in the dark. The calibration curve was made from a stock solution of 10 mM Trolox in 70% ethanol, with a concentration interval of 0.25–2 mM Trolox.

### 2.6. Evaluation of Interaction between Honey/GFSw and Honeysuckle Extract in Terms of AOA 

In order to evaluate the interaction between honey/GFSw and honeysuckle extract (H_GFSw_LFL) in terms of AOA and also to compare it to the AOA behaviour of H/GFSw_RE, we first used the procedure described in our previous research [5]. This procedure involved the determination of the combination index, isobolograms, the dose–response curve of each compound, and the evaluation of theoretical and experimental AOA of the samples. The combination index (CI) of mixture H_LFL/GFSw_LFL was calculated based on the ratio of the concentration of each compound when combined in the mixture(*Cc*_1,*c*_ and *Cc*_2,*c*_) to the concentration when used separately (*Cc*_1,*s*_ and *Cc*_2,*s*_) to achieve the same effect as observed in the mixture [31]:$$CI=\frac{C_{c1,\;c}}{C_{c1,s}}+\frac{C_{c2,c}}{C_{c2,\;s}}$$
where *Cc*_1_ means the concentration of H/GFSW, and *Cc*_2_—is the concentration of extract of honeysuckle (LFL). Isobolomic analysis was the graphical representation of the same data. 

The CI and isobolomic analysis of the samples in the case of FRAP and CUPRAC methods was expressed as the effective concentration of the samples at 1 mM Trolox (EC 1 mM Trolox, mg/mL).

In the case of ABTS and DPPH methods of AOA, for evaluation of the CI and isobolograms of the samples, the values of IC_50_ and IC_20_ (50% and 20% inhibitory concentration of the substrate) were used. The IC_50_ and IC_20_ values were calculated based on the median-effect equation, transforming the non-linear equation for the dose–response curve into a linear one:$$log\frac{f_{i}}{f_{u}}=m\times log\left(conc.\right)+n$$

*f_i_* and *f_u_* are inhibited and uninhibited fractions of the reaction, *m*—the slope and *n*—respectively intercept of the curve.

*f_u_*—is inhibited fraction of the substrate by antioxidant sample and was calculated as follows:$$f_{i}=\frac{\left(\;A_{0}-A_{blank\_0}\right)-\left(A_{s}-A_{blank\_s}\right)}{\left(A_{0}-A_{blank_{0}}\right)}\times 100\%$$
where $A_{0}$ is the absorbance of the substrate (DPPH or ABTS reagent), $A_{blank\_0}$—absorbance of the blank of the substrate (solvent), $A_{s}$—absorbance of the sample, $A_{blank\_s}$—absorbance of the blank of the sample (sample without substrate).

*f_u_*—is the uninhibited fraction of the reaction and was calculated as *f_u_ =* 100 − *f_i_.*

Another way to evaluate the interactions between polyphenols and honey/GFSw was by plotting concentration–dependent curves of the theoretical and experimental antioxidant activities. In the case of FRAP and CUPRAC methods, the theoretical AOA was calculated by addition (absorbance of H/GFSw + absorbance of LFL). In the case of ABTS and DPPH methods, which were non-linear dose–response curves, the theoretical AOA of the samples was calculated by the Webb equation: 100 − ((100 − *f_i,C_*_1_) × (100 *− f_i,C_*_2_)), where *f_i,C_*_1_ and *f_i,C_*_2_ are the inhibited fraction of compound 1 (C1) and compound 2 (C2), respectively, when analysed separately.

For comparison of LFL and RE within the experimental concentration range, we generated dose–response curves, Dose–Response Matrix, and 2D representation of Synergy Score for DPPH and ABTS using the SynergyFinder R package [32]. We conducted an analysis of the interactions between the components within H_ LFL, GFSw_LFL mixtures from this study, and H_RE and GFSw_RE from our previous study [5]. We generated Dose–response curves for LFL and RE in the absence and presence of honey/GFSw, as well as for honey and GFSw. The common concentration interval and concentration values between LFL and RE were chosen. The objective of this analysis was to compare the degree of inhibition of DPPH and ABTS radicals, both the experimental and those simulated by SynergyFinder. However, the FRAP and CUPRAC methods could not be analysed through the SynergyFinder R package as these methods do not result in an inhibition percent.

### 2.7. Prebiotic Activity

The prebiotic activity of the mixtures of honey/GFSw enriched with dried plant extract (raspberry and honeysuckle flowers) was assessed by the evaluation of the growth-promoting activity of the samples on the strain of *Limosilactobacillus reuteri* DSM 20016 and determination of L-lactic acid content as a metabolite produced during of sample fermentation. The results were compared to simple honey/GFSw, plant extract at the same concentration as in the mixture, and the control (C+) of strain. The statistics were conducted between extract (RE, LFL) and C+, and between honey/GFSw mixtures and simple honey/GFSw.

#### 2.7.1. Probiotic Growth-Promoting

The evaluation of the growth-promoting of the samples was performed according to the method described by [33,34] with some modifications. The probiotic strain of *L. reuteri* DSM 20016 was obtained from Leibniz Institute DSMZ-German Collection of Microorganisms and Cell Cultures GmbH (DSMZ, Braunschweig, Germany). The strain was stored in a cryotube with 25% (*v*/*v*) glycerol solution at −90 °C. Before experiments, the probiotic was activated by being inoculated in MRS broth for 48 h at 30 °C in Oxoid™ AnaeroJar™ 2.5 L (Thermo Scientific™) and after was subcultured on MRS agar plate under the same conditions for preparing the probiotic inoculum (0.5 McFarland) in sterile saline solution (0.8% NaCl).

The stock solution of the samples (50 mg/mL for H/GFSw and their mixtures and 2.25 mg/mL for LFL and RE) was prepared by solubilisation in MRS broth and sterile filtration through sterile 0.22 µm PES filters. The test itself was carried out in Eppendorf tubes by making dilutions in MRS to obtain 5 test concentrations between 1–45 mg/mL (for H/GFSw and their mixtures) and 0.05–2.25 mg/mL for plant extract (LFL and RE). The concentration of the samples was calculated for the final volume in the test tube after adding 10% of probiotic inoculum. The control sample (C+, which means the control of the strain *L. reuteri* without any supplements) was prepared in the same way as the samples by adding 10% of probiotic inoculum in the medium MRS broth. The samples were incubated for 48 h at 30 °C in Oxoid™ AnaeroJar™, and the absorbance of the samples was measured at 600 nm in 96-well plates using a plate reader after carefully and thoroughly mixing the Eppendorf tube. The samples were stored in the freezer at −20 °C for further analysis of lactic acid content.

The growth-promoting effect of the samples was calculated as follows: (A_s_ − A_blank_s_)/A_c_ − A_blank_c_) × 100, where A_s_—absorbance of the sample after incubation time, A_c_—absorbance of control samples of the strain, A_blank_s_—absorbance of the blank of the sample (before incubated time), A_blank_c_—absorbance of the blank of the control of the strain before incubated time. The results are expressed as percent bacterial growth.

#### 2.7.2. L-Lactic Acid Content

The L-lactic acid content produced during cultivation of *L. reuteri* in the presence of the samples tested was determined enzymatically using the commercial kit–K-DLATE kit for D-/L-Lactic Acid (D-/L-Lactate) (Rapid) Assay Kit (Megazyme, International Ireland Ltd., Wicklow, Ireland). Before analysis, the samples were centrifugated at 1470 rcf for 10 min, and the supernatant of the samples was analysed according to the manufacturer’s kit protocol.

### 2.8. Statistical Analysis

Statistical analysis for prebiotic activity was performed using IBM ^®^SPSS^®^ Statistics, version 26 (IBM SPSS Corp., Armonk, NY, USA). All assays were carried out in triplicate, and the results are expressed as mean values ± standard deviation (SD). One-way analysis of variance (ANOVA) was used to determine if significant differences exist between the tested samples of honey and GFSw with or without plant extract (honeysuckle and raspberry extract) and vice-versa. The homogeneity of variance was tested by Levene’s test. To explore the significant difference between group means, Tukey’s honestly significant difference (HSD) test was performed for homoscedastic groups and Games–Howell for heteroscedastic groups.

For the isobolographic analysis of the AOA, we computed 95% confidence intervals. These intervals were determined by subtracting and adding the value of 1.96 times the standard deviation (SD) divided by the square root of the number of measurement replicates (n) from the mean of results (mean ± 1.96 × SD/sqrt (n)). In this study, the number of replicates was three (n = 3) for all cases.

## 3. Results

### 3.1. Screening of Polyphenolic Compounds from the Honeysuckle Flower

The polyphenolic profile of honeysuckle flowers was evaluated by several methods: total polyphenols content (TPC), total flavonoids content (TFC), total hydroxycinnamic acid content (HAT), total anthocyanin content (TAC), and HPLC assays and the results are summarized in Table 1 and Table 2.

The phenolic acids and flavonoids found and identified in honeysuckle flowers by HPLC analysis after ultrasound-assisted extraction were caffeic acid (RT–23.62 min), chlorogenic acid (RT–22.50 min), ferulic acid (RT–30.68 min), p-coumaric acid (RT–33.91 min), epicatechin (RT–14.61 min) and apigenin (RT–17.20 min) The chromatograms illustrating the polyphenolic compounds found in honeysuckle flowers can be found in the Appendix A. Caffeic acid, 36.54 ± 0.04 mg/g DW, and epicatechin, 2.83 ± 0.02 mg/g DW had the highest content among polyphenols.

### 3.2. Evaluation of the Interaction between Honey/GFSw and Extract of Honeysuckle Flowers in Terms of AOA

The honey mixture with honeysuckle extract (H_LFL) showed much higher antioxidant activity (AOA) compared to commercial honey at all concentrations tested (2–200 mg/mL) as determined by all the methods of AOA (FRAP, CUPRAC, DPPH, ABTS) evaluated. Appendix A provides a detailed comparison of the AOA values of the analysed samples, illustrated by the dose–response curves for the AOA of the sample. We can see that the slope and intercept values of H_LFL were much higher than the values for simple honey. Similar behaviour can be observed for the mixture between GFSw and LFL (GFSw_LFL). 

In an intention to evaluate the interactions between LFL and honey/GFSw and to compare them with our previous data reported for the raspberry extract [5], the combination index (CI) was calculated (Table 3) based on the calibration curves of the samples, and the isobolograms were plotted—Figure 1 DPPH and ABTS) and Figure 2 (FRAP and CUPRAC). 

In accordance with our previous study [5], we have categorized the CI values for ease of comparison as follows: 0.5–0.7 indicates strong synergism, 0.7–0.9 denotes moderate synergism, 0.9–1.1 implies nearly additive behaviour, 1.1–1.5 signifies moderate antagonism, 1.5–2 indicates moderate to strong antagonism, and CI > 2 represents strong antagonism.

The interaction between LFL and honey ranged from moderate synergism (CI = 0.86 ± 0.04 for FRAP) to nearly additive behaviour (CI = 1.03 ± 0.02 for ABTS IC50 and 1.03 ± 0.01 for ABTS IC20) and moderate antagonism (CI = 1.16 ± 0.04 for DPPH IC50 and 1.16 ± 0.04 for DPPH IC20, CI = 1.29 ± 0.020 for CUPRAC). GFSw with LFL exhibited similar behaviour in the case of FRAP (CI = 0.85 ± 0.03), ABTS IC50 and ABTS IC20 (CI = 1.09 ± 0.04 and 1.09 ± 0.02, respectively), and CUPRAC (CI = 1.35 ± 0.19). However, some differences in behaviour were observed in terms of DPPH, the CI being higher than in the case of H_LFL. The DPPH CI values of GFSw_LFL indicated a moderate antagonism feature (CI = 1.35 ± 0.19 for DPPH IC50) and moderate to strong antagonism (CI = 1.54 ± 0.07 for DPPH IC20). 

The behaviour of H_LFL is very similar to the behaviour of GFSw_LFL in terms of CI and ranges from synergism, additive, and antagonism (0.85 ± 0.029 FRAP for GFSw_LFL and 1.536 ± 0.071 DPPH IC20 for GFSw_LFL). The only synergic behaviour was obtained in the case of FRAP.

The isobologram is a graphical representation of the interactions between two compounds of a mixture and represents the effect of separate compounds when they are in a mixture.

The x- and y-axes on the graph represent the concentrations of the compounds in the mixture, specifically honey (H)/GFSw and honeysuckle extract (LFL). In Figure 1, the black and green circles on the graph represent the effects of IC_50_ (half-maximal inhibitory concentration) and, respectively, IC_20_ (20% inhibitory concentration) of H/GFSw and LFL when each of the two components is used individually. Before analyzing the interactions between the compounds graphically, an additive line is drawn between the two compounds (between the black or green circles). The concentrations of the components in the mixture that gave the same result (IC_50_, IC_20_) were plotted as stars. If the position of the mixture lies above the additive line, it indicates antagonism. If it lies below the additive line, it indicates synergism. If it lies on the additive line, it represents an additive effect.

As can be observed in Figure 1, the mixtures of honey with LFL and GFSw with LFL exhibited similar behaviour. Furthermore, no significant difference between the behaviour of the mixture at IC_50_ and IC_20_ was observed, and the data corroborated with the CI results from Table 3.

Figure 2 illustrates the isobolograms of the correlation between honey (H)/GFSw and LFL regarding AOA using the FRAP and CUPRAC methods. It is evident from the figure that the mixture of honey and GFSw with LFL displayed comparable behaviour, which correlated with the CI data from Table 3.

To further assess the interactions between polyphenols and honey/GFSw, concentration–dependent curves of both theoretical and experimental AOA (Figure 3 and Figure 4) were plotted. This provided an additional means of evaluating the extent and nature of these interactions. 

As previously stated, the theoretical AOA for the FRAP and CUPRAC methods was obtained by summing the absorbance values of H/GFSw and LFL. However, for the ABTS and DPPH methods, which exhibited nonlinear concentration–dependent curves, the theoretical AOA was calculated using the Webb equation, taking into account the inhibited and uninhibited fractions.

By examining the relative positioning of the concentration-dependence curves for the theoretical versus the experimental AOA of the analyzed samples, we can assess the modulation of polyphenols in honey and GFSw. If these curves overlap, it indicates an additive effect between the components. If the theoretical (calculated) curve lies below the experimental curve, it suggests synergism, whereas if the theoretical curve is higher than the experimental curve, it indicates antagonism.

The observed trend indicates that the AOA exhibited a linear relationship with concentration for the FRAP and CUPRAC methods, and a sigmoidal relationship was observed for the DPPH and ABTS methods.

For a more in-depth comparison between LFL and RE, we analysed the DPPH and ABTS data using SynergyFinder R package. The dose–response Curves from Appendix A confirm that LFL has a higher AOA than RE. The curves of LFL start to saturate at the maximum concentration tested, with a final inhibition of approx. 80%, but the curves of RE are still on the ascendent trend, reaching approx. 50–60% inhibition at the same concentration. We showed in our previous work [5] that the saturation takes place at RE mixture concentrations higher than 100 mg/mL (5 mg/mL RE extract). 

Based on the dose–response curves of the sample analysed (in the case of samples with RE, the dose–response curves based on the data from our previous article [5] were used), the values of IC_50_ (for DPPH and ABTS methods) and Trolox Equivalent Antioxidant Capacity (TEAC coefficient) were calculated and are shown in Table 4 (from the AOA as a function of extract concentration in the mixture) and Table 5 (from the AOA as a function of total mixture concentration). 

The IC_50_ value is a measure of the concentration of a substance needed to inhibit a specific biological or biochemical activity by 50%. In the context of antioxidant activity, a lower IC_50_ value indicates a stronger antioxidant capacity, as it implies that a lower concentration of the sample is required to achieve the same inhibitory effect. 

The results in Table 4 show that LFL had an almost 10× lower IC50 and higher TEAC than RE, both applied alone or in a mixture. Honeysuckle flower extract demonstrated consistently superior AOA performance compared to raspberry extract in all methodologies examined. Honey improved the AOA only in the case of DPPH and FRAP of RE, which correlates with the synergism reported previously [5], and FRAP of LFL, which correlates with the small synergism observed at higher concentrations for FRAP of H_LFL in Figure 4. GFSw improved the AOA only in the case of FRAP of LFL, which correlated with the small synergism in Figure 4. All the other mixtures showed similar or lower AOA compared with the extract itself. In the case of RE, the AOA was higher in mixtures with honey than with GFSw, except for CUPRAC, where the AOA is the same. In the case of LFL, honey and GFSw behaved similarly.

A similar trend as in Table 4 is observed when the total mixture concentration is used (Table 5). 

The Dose–Response Matrix (DRM) and Loewe Synergy score (LSS) for DPPH generated by SynergyFinder are shown in Figure 5 and Figure 6. The points that form the diagonal represent the experimental points, and the rest of the combinations represent predicted behaviour generated by the software. The following observations can be drawn from DRM: all DRM have very similar patterns, except H_RE and H_CA, which present more significant differences and look similar one to the other; the H_LFL and GFSw_LFL are almost identical; the earlier saturation and higher AOA of H_LFL compared to H_RE is also predicted for other combinations of concentrations. 

The LSS confirms the prevalence of synergism for RE and antagonism for LFL for diagonal combinations and predicts the same difference in other combinations. The synergism pattern of H_LFL and GFSw_LFL are again almost identical, while there are some differences in the case of RE.

In the case of ABTS, a similar difference between LFL and RE is observed in DRM (Figure 7). In this case, H_RE resembles not only H_CA but also H_EP, and GFSw induces changes that give patterns similar to that of H_LFL and GFSw_LFL, which resemble very much, just as in the case of DPPH. 

The LSS of ABTS confirms the predominant antagonist behavior for the experimental points (diagonal) and predicts similar behavior at other concentrations (Figure 8).

### 3.3. Prebiotic Activity

The prebiotic activity of the mixtures of honey (H)/GFSw enriched with dried plant extracts (raspberry fruits and honeysuckle flowers) was evaluated by assessing their growth-promoting effects on the *Limosilactobacillus reuteri* DSM 20,016 strain and measuring the production of L-lactic acid as a metabolite during sample fermentation. In order to assess whether the improved antioxidant activity of honey also improves the prebiotic activity, five concentrations ranging from 45–1 mg/mL of H/GFSw and their mixtures with the two extracts (LFL and RE) were tested. The individual extracts were tested at concentrations of 2.25–0.05 mg/mL corresponding to their respective concentrations in the mixtures—Figure 9.

As observed in Figure 9, the majority of samples exhibited a positive impact on bacterial growth compared to the control (C+, which is the control sample of the strain *L. reuteri* without any supplements), except GFSw at lower concentrations (1 mg/mL–96.76 ± 1.93%) and the LFL extract at low concentrations (1 mg/mL–97.29 ± 0.94%, and 5 mg/mL–97.63 ± 1.77%). Moreover, the highest bacterial growth was observed for the mixture of honey/GFSw with LFL compared to simple honey/GFSw, particularly at a concentration of 25 mg/mL (statistically significant differences with a σ-value of 0.000). The growth percentages were 145.38 ± 0.29% for H_LFL compared to 134.08 ± 3.10% for simple honey, and 154.69 ± 0.13% for GFSw_LFL compared to 143.38 ± 2.36% for GFSw. Furthermore, GFSw samples exhibited a slightly stronger influence on the growth of the *L. reuteri* strain compared to honey at all tested concentrations, except for the lowest concentration tested of 1 mg/mL, but all honey concentrations showed prebiotic activity. GFSw at the lowest concentration tested of 1 mg/mL had a slight inhibitory effect. The growth percentages from the highest to the lowest concentration tested were 160.02% to 96.76% for GFSw and 158.08% to 102.96% for honey. 

At the highest concentration (45 mg/mL), the mixture of honey with raspberry extract (RE) demonstrated a lower prebiotic activity in terms of bacterial growth (149.93 ± 0.12%) when compared to simple honey (158.08 ± 2.19%). Conversely, slightly higher probiotic growth is observed for the mixture of honey/GFSw with RE compared to simple honey/GFSw, at 25 mg/mL (137.29 ± 1.98 for H_RE in comparison with 134.08 ± 3.09% for simple honey and 147.32 ± 3.53 compared to 143.38 ± 2.36% for GFSw), but the differences are not statistically significant. Overall, RE does not induce a prebiotic effect neither in the absence nor in the presence of honey/GFSw at the concentrations tested.

According to our data, honey mixtures with honeysuckle extracts exert a slightly more positive effect on bacterial growth than those with raspberry extract, the results being in the range of 164.63 ± 0.12–104.13 ± 0.26% for H_LFL and 149.93 ± 0.12–102.94 ± 1.43% for H_RE. Similar data were observed in the case of the GFSw mixture, the value of bacterial growth being in the range of 161.514 ± 0.17–100.55 ± 0.32% for GFSw_LFL and 157.47 ± 1.42–101.67 ± 0.82% for GFSw_RE. 

Upon observation, it is evident that both the mixtures and individual samples of honey and GFSw exhibit concentration–dependent effects on the probiotic growth, where the observed effect diminishes as the concentration decreases.

Regarding L-lactic acid production, the effects are diverse due to the complex metabolic interactions between polyphenols and the carbohydrate metabolism in heterofermentative lactic bacteria (like the used *L. reuteri* DSM 20016) under anaerobic conditions (Figure 9C,D). The individual extracts, LFL and RE, behaved relatively differently. The maximum positive effect of LFL compared to C+ was at the median LFL concentration of 10 mg/mL (1.85 ± 0.00 g/L versus 0.86 ± 0.02 g/L L-lactic acid, respectively), followed by 25 mg/mL LFL (1.22 ± 0.03 g/L L-lactic acid). Both values were statistically significant. At the lowest LFL concentration tested, 1 mg/mL, there was a significant inhibition of L-lactic acid production (0.19 ± 0.05 g/L) compared to C+. The other two LFL concentrations, 45 and 5 mg/mL, did not have a significant effect compared to C+ (Figure 9C). 

Most of the RE concentrations tested had a positive effect on the L-lactic acid production except the lowest RE concentration, 1 mg/mL, which had a slight but not statistically significant inhibitory effect (0.72 ± 0.03 g/L L-lactic acid) compared to control C+. The only statistically significant positive effect compared to C+ was at the highest RE concentration of 45 mg/mL (1.47 ± 0.02 g/L L-lactic acid).

A statistically significant difference was observed in the L-lactic acid content between H_LFL and simple honey at tested concentrations of 45 mg/mL, 10 mg/mL, and 1 mg/mL (Figure 9C). The first two concentrations exhibited a positive trend, with higher L-lactic acid content (2.96 ± 0.09 g/L at 45 mg/mL of H_LFL and 2.22 ± 0.04 g/L at 10 mg/mL of H_LFL) compared to simple honey fermentation (1.60 ± 0.03 g/L L-lactic acid at 45 mg/mL of H, and 1.52 ± 0.03 g/L L-lactic acid at 10 mg/mL of H). At the concentration of 1 mg/mL H_LFL, the L-lactic acid content was lower (0.75 ± 0.03 g/L) than that observed during H fermentation (2.36 ± 0.02 g/L).

In the case of the mixture of honey with raspberry extract, higher L-lactic acid content than in the case of simple honey was observed at the tested concentrations 10 and 5 mg/mL (2.01 ± 0.08 g/L for H_RE and 1.52 ± 0.03 g/L for H at 10 mg/mL, 1.93 ± 0.05 g/L for H_RE and 1.26 ± 0.01 g/L for H at 5 mg/mL).

According to our data, the highest positive effect of GFSw_LFL on the L-lactic acid content produced by *L. reuteri* compared to simple GFSw was at 10 mg/mL (2.22 ± 0.12 g/L L-lactic acid for GFSw_LFL compared to 1.93 ± 0.09 g/L L-lactic acid for GSFw). Other statistically significant positive effects on the L-lactic acid content were obtained at 25 and 1 mg/mL GFSw_LFL compared to GFSw. At the highest GFSw_LFL concentration tested, 45 mg/mL, there was an inhibition of L-lactic acid production (Figure 9C). The highest effect of GFSw_RE in comparison to GFSw was at 25 mg/mL tested concentration (2.24 ± 0.003 g/L for GFSw_RE and 1.18 ± 0.092 g/L for GFSw). The only GFSw_RE concentration that inhibited the production of L-lactic acid was the highest concentration of the mixture, 45 mg/mL (Figure 9D).

## 4. Discussion

Based on the Information found in previous studies [35,36,37,38], the concentration of phenolic compounds is influenced by various factors such as plant species, cultivars, environmental conditions, storage, extraction methods, and analysis techniques. Consequently, the reported concentrations of phenolic compounds can differ significantly across different scientific articles, and it is difficult to make direct comparisons. In our study, the value of TPC from honeysuckle was in the range of some literature data [21,36,38]. The TPC value was lower compared to the results reported by [38]–87.48 ± 6.32 mg GAE/g and by other researchers [36], who extracted the polyphenols in water (40.18 mg GAE/g) and in ethanol (5.25 mg GAE/g). The TPC values were higher compared to those obtained by our research group (392.093–1741.05 µg GAE/g DW) in another study [21]. This is due to repeated extraction from the same substrate. The value of TFC in our study was much lower than those revealed by [38] –52.51 mg CAE/g, but they expressed the TFC results as catechine equivalent (CAE) in comparison to our result (QE—quercetin equivalent). We tested catechine at the same concentrations as quercetin and found it to have much lower activity than quercetin; therefore, more catechin is needed to have the same AOA as 1 quercetin equivalent, which could explain the difference. Moreover, [38] employed a different methodology to determine the flavonoid content, involving the use of AlCl_3_, NaNO_2_, and NaOH. Additionally, the extraction of phenolic compounds in their study was conducted using a solution containing acetone, water, and acetic acid (70:29.5:0.5, *v*/*v*/*v*). 

The results of TPC, TFC, and HAT (651.79 ± 5.11 GAE mg/100 g DW, 64.56 ± 2.12 QE mg/100 g DW, 587.38 ± 1.19 ChaE mg/100 g DW) were substantially higher in comparison to the raspberry extract (282 ± 10.72 GAE mg/100 g DW, 29.88 ± 1.05 QE mg/g DW, and 57.92 ± 2.92 Chae mg/100 g DW) from our previous results [5].

Chlorogenic acid and caffeic acid have also been identified in honeysuckle flowers by other authors [35,36,37]. The content of chlorogenic acid was lower than the values obtained by [35]–33.12 ± 0.25–48.84 ± 0. 04 µg/mg (depending on the growth stage of flowers), and our value was higher compared to the results reported by [21]—1331 µg/g. The content of caffeic acid in our case was lower than the results released by [35] 0.01–0.07 µg/mg for *L. japonica* and [39], which obtained a value of 0.195 ± 0.002 g/100 g. As can be seen, the values differ very much between studies.

As mentioned previously, the objective of this study was to build upon prior research by exploring the impact of incorporating a different polyphenol extract into honey or its biomimetic natural deep eutectic solvent (NaDES) and comparing it with the AOA behaviour of mixtures enriched with the raspberry extract from our previous study [5].

There is a correlation between the antioxidant activity of the samples (honey and plant extract) and the concentrations and profiles of the analyzed polyphenols, as determined by colourimetric tests (TPC, TFC, HAT) and HPLC analysis. In the case of the samples enriched with honeysuckle flower extract, the antioxidant activity (AOA) had to be analyzed at lower concentrations, ranging from 2 to 100 mg/mL, and the AOA of samples enriched with raspberry extract was analyzed at concentrations ranging from 5 to 200 mg/mL. 

Additionally, in some cases, the samples with *L. caprifolium* extract exceed the detection limit of the method and instrument (absorbance at 200 mg/mL exceeds 3.5)–particularly in the case of the FRAP method, for which additional optimization of the method will be necessary in the future, such as the decrease of substrate concentration. Furthermore, at elevated concentrations (200 mg/mL for H_LFL/GFSw_LFL or 10 mg/mL for LFL), the dose–response curves for FRAP and CUPRAC methods deviated from linearity. Also, the AOA of the sample with LFL shows inhibition at 200 mg/mL when measured using the ABTS and DPPH methods (Appendix A).

These issues can be attributed to the higher concentrations of active compounds extracted from *L. caprifolium* flowers compared to raspberry, as also observed in the TPC, HAT, TFC, and HPLC analyses. 

As observed in Table 4, there are differences in terms of antioxidant activity of samples enriched with raspberry extract and samples enriched with honeysuckle flower extract. The samples with honeysuckle extract showed higher AOA than those with raspberry extract. We can conclude that there is a relationship between the polyphenol content and antioxidant activity, as the polyphenol content was higher in LFL than in RE by all the methods tested.

As mentioned previously, the incorporation of dried honeysuckle flower extract (LFL) into honey or its biomimetic NaDES, GFSw, increased the antioxidant activity of both honey and GFSw. To evaluate the modulation behaviour of LFL by honey and GFSw, the combination index (CI) was calculated. 

The two extracts share some similar main compounds (caffeic acid–CA and epicatechin–EP), but they are different in the total polyphenolic content. In our previous study, we investigated the individual CA and EP as well [5]. It seems that the behaviour of the honeysuckle extract is similar to the behaviour of these tested polyphenols in certain cases. For example, in the case of the FRAP method, the AOA feature of LFL in honey was similar to CA in honey (CI = 0.866 ± 0.021), both of them exhibiting moderate synergistic effects. This similarity between LFL and CA is also observed for the CUPRAC and DPPH IC_50_ methods, the AOA behaviour of CA being moderate antagonism with CI value 1.43 ± 0.02 for CUPRAC and 1.17 ± 0.08 for DPPH IC_50_. In the case of epicatechin and its behaviour in honey or GFSw, a similarity to LFL is observed in the case of DPPH IC_20_ (CI = 1.4 ± 0.11). The similarities in terms of antioxidant activity between LFL and CA are likely attributed to the higher content of CA in LFL (36.54 ± 0.04 mg/g DW)).

In comparison to raspberry extract, which exhibited varying behaviour depending on the tested concentrations, the AOA behaviour of honeysuckle extract demonstrated minimal variation. This can be observed in DPPH and ABTS assays at both 50% and 20% inhibition substrate, where the combination index (CI) values were nearly identical.

The interactions between LFL and GFSw are similar to RE and GFSw interactions in the case of the CUPRAC method, RE exerting moderate antagonistic behaviour (CI = 1.409 ± 0.023) and ABTS IC_20_ with nearly additive feature (CI = 1.011 ± 0.079).

In accordance with our previous study [5], we codified the CI intervals as follows: 0.5–0.7, which indicates strong synergism as (+2), 0.7–0.9, which denotes moderate synergism as (+1), 0.9–1.1 which implies nearly additive behaviour as (0), 1.1–1.5, which signifies moderate antagonism as (−1), 1.5–2, which indicates moderate to strong antagonism as (−2), and CI > 2, which represents strong antagonism as (−3).

Figure 10 reveals that the antioxidant behaviour (AOA) of polyphenols, including the two plant extracts and the polyphenol standards, in honey and GFSw exhibited a similar tendency, as indicated by a similar colour code. The strong antagonism values (−3) are assigned to dark blue and strong synergism(+2) to light blue. Out of the total 24 cases analyzed, 14 cases, accounting for approximately 58%, demonstrated that honey and GFSw (biomimetic NaDES) behaved similarly. From Figure 10, it is easy to see that LFL induced a more homogeneous behaviour than RE. It is also suggested that, in fact, the AOA of the extracts is a result of the combined effects of different polyphenolic species, to which inter-polyphenolic interactions probably contribute. The quantitative values of the qualitative representation from Figure 10 can be found in Appendix A.

Honey and polyphenols are both well-known for their prebiotic properties [14,40,41], which means they can support the growth and activity of beneficial bacteria in the gut. The prebiotic properties of honey are attributed to the presence of oligosaccharides, short-chain carbohydrates that are not fully digested in the upper gastrointestinal tract. Instead, they reach the colon intact, where they can exert prebiotic effects. Despite honey being primarily composed of simple sugars that are quickly absorbed in the small intestine, there are also di-, tri-, and oligosaccharides present in smaller quantities. These low-weight polysaccharides are likely to resist degradation by host enzymes, allowing them to reach the lower gut and contribute to the prebiotic effects of honey [13,14,42]. There are numerous studies on the prebiotic activity of honey [13,34,43,44,45]. In most cases, honey exerted positive effects on probiotic growth, and the prebiotic activity of the analysed substrate is often influenced by the concentration of substances they are exposed to. For example, in the study reported by [34], they tested two concentrations (1% and 2%) of several types of honey on five probiotic strains (*Lactobacillus acidophilus*, *Lactiplantibacillus plantarum*, *Lactobacillus gasseri*, *Lacticaseibacillus rhamnosus*, and *Lacticaseibacillus casei*,), the highest prebiotic activity expressed in terms of bacterial growth was observed at 2% honey in the case of *L. plantarum* (6× greater than control) and *L. acidophilus* (4× higher than control). In another study [45], different levels of active Manuka Factor (AMF:0.5, 10, 15, 20) were tested on the growth of the *strain Limosilactobacillus reuteri* DPC16, and it was observed that the highest biomass of probiotic substrate was obtained at AMF20 (4.77 mg/mL) after 36 h of incubation under anaerobic conditions compared to control (2.23 mg/mL).

In the context of our study, where honey and mixtures of honey/GFSw enriched with dried plant extracts were tested for prebiotic activity, it was observed that almost all concentrations of these mixtures had a positive impact on the growth of the probiotic strain *Limosilactobacillus reuteri* DSM 20016. Only the lowest concentration tested, 1 mg/mL did not show prebiotic activity. The higher the concentration of honey or the mixtures, the greater the positive effect on the probiotic growth. The data show that the prebiotic effects manifest after a certain concentration of compounds.

The prebiotic effect was mainly induced by honey, so probably by the saccharides present in it. The extracts showed a moderate prebiotic effect, statistically significant only in the case of LFL at the highest concentrations tested. As the higher the LFL extract, the higher the prebiotic effect, higher extract concentrations should be tested until reaching a plateau or an inhibition.

Based on our findings, the honey/GFS_W_ mixtures enriched with honeysuckle or raspberry did not show significant differences at most of the tested concentrations. The most significant difference was at the tested concentration of 45 mg/mL; the honey mixture with LFL exhibited a more pronounced effect (164.63 ± 0.01%) compared to the honey and RE mixture (149.93 ± 0.12%) on the bacterial growth (*p*-value = 0.019, <0.05). H_LFL had a prebiotic effect, and H_RE had an inhibitory effect compared to H. The higher content of hydroxycinnamic acids and lower anthocyanin content that we determined in the LFL extract compared to RE [5] could be involved in this effect, but more studies are needed to understand the mechanism behind this difference. 

Most concentrations of honey and GFSw had positive effects on the production of L-lactic acid, as the lactic acid bacteria (LAB) metabolize sugars into lactic acid. The dependence of H/GFSw concentration presented an apparent hormetic behaviour nevertheless. In the case of the extracts, the maximum effect of LFL on the L-lactic acid content at lower concentrations than RE is probably related to the higher polyphenols content of LFL compared to RE. The mixtures presented a less-evident hormetic effect than H/GFSw, and there are two differences worth mentioning: (1) the trend of H_LFL is opposite (increasing effect with the concentration) to the trend of H_RE (decreasing effect with the concentration); (2) both GFSw_LFL and GFSw_RE differ from the corresponding honey mixtures, the first having the maximum effect at intermediate concentration tested (10 and 25 mg/mL, respectively). This suggests that honey interacts differently with LFL and RE, probably due to other compounds present in honey than saccharides.

Hydroxycinnamic acids are used as external electron acceptors by heterofermentative lactic acid bacteria [46], therefore decreasing the production of lactic acid, that is, the product of NAD(P)H reoxidation by using pyruvic acid as external electron acceptors [47]. However, other polyphenols exert different effects on lactic acid bacteria being metabolysed by several different enzyme classes besides the reductases, e.g., esterase and decarboxylases used for hydroxycinnamic acids [48]. The complex effects of polyphenols on lactic acid production require more investigation.

## 5. Conclusions

Although honey/GFSw modulated the AOA of both extracts, from synergism to antagonism, the modulation was different between the two extracts for some AOA activities, which could be explained by the differences between compositions in polyphenols of the two tested plant extracts. The effects are specific to complex chemical systems, wherein the biological and biochemical activities are determined by multiple interactions. The honeysuckle flower extract (LFL) has higher prebiotic activity than the raspberry extract. The effect on lactic acid production follows a hormetic behavior. 

## Figures and Tables

**Figure 1 antioxidants-12-01678-f001:**
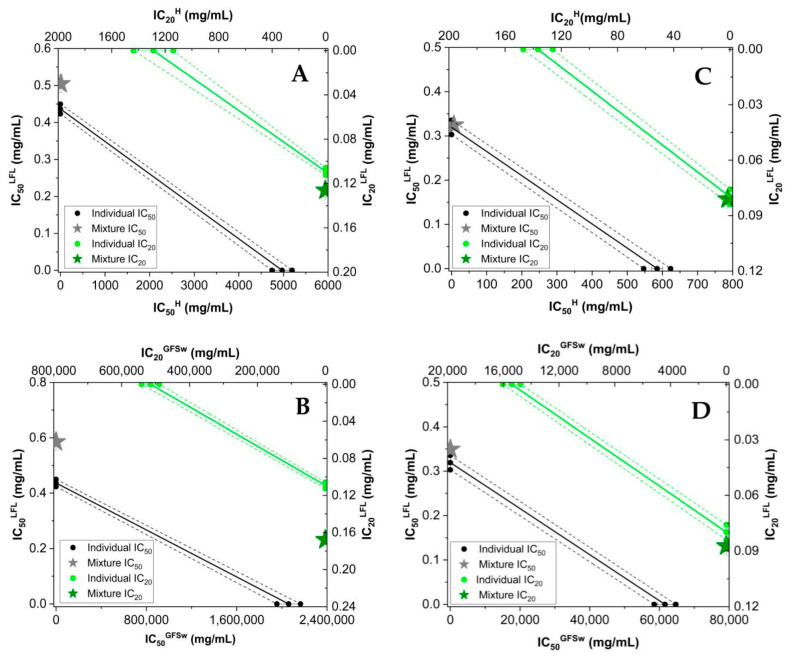
Isobolograms of honey (H) and its biomimetic NaDES (GFSw), and honeysuckle flowers extract (LFL) in terms of AOA by DPPH (**A**,**B**), and ABTS (**C**,**D**); IC_50_ (half-maximal inhibitory concentration) and IC_20_ (inhibitory concentration at 20% substrate inhibition); dashed lines indicate the 95% confidence intervals.

**Figure 2 antioxidants-12-01678-f002:**
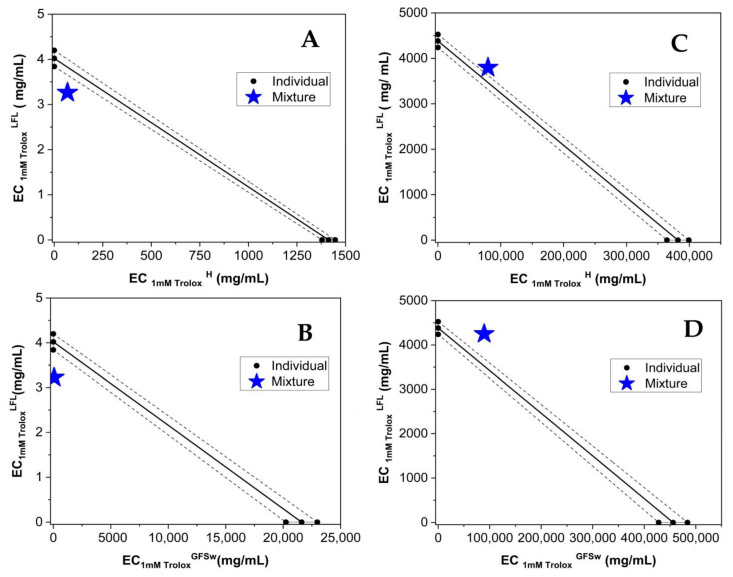
Isobolograms of honey (H) and its biomimetic NaDES (GFSw), and honeysuckle flowers extract (LFL) in the therm of AOA by FRAP (**A**,**B**) and CUPRAC (**C**,**D**) methods, CE- effective concentration at 1 mM Trolox equivalent of the sample. Each value is accompanied by error bars representing three measurements. Dashed lines indicate the 95% confidence intervals.

**Figure 3 antioxidants-12-01678-f003:**
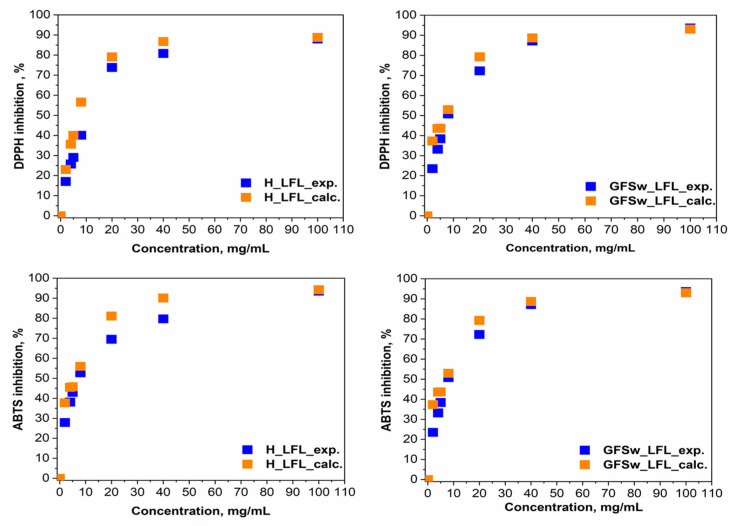
Webb analysis of experimental (H_LFL/GFSw_LFL_exp.) and theoretical (H_LFL/GFSw_LFL_calc.) AOA in the mixture of honey (H)/GFSw with the honeysuckle extract (LFL).

**Figure 4 antioxidants-12-01678-f004:**
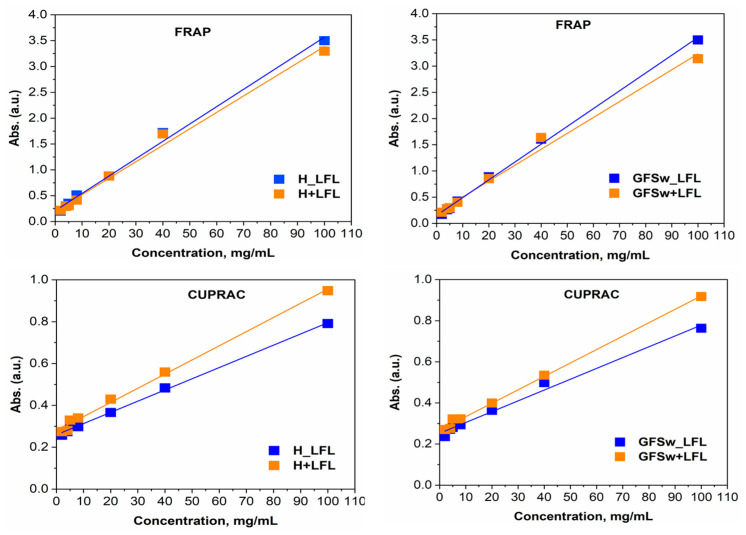
Evaluation of the concentration dependence of experimental (H_LFL/GFSw_LFL) and theoretical (H+LFL/GFSw + LFL) AOA in the mixture of honey(H)/GFSw with the honeysuckle extract (LFL).

**Figure 5 antioxidants-12-01678-f005:**
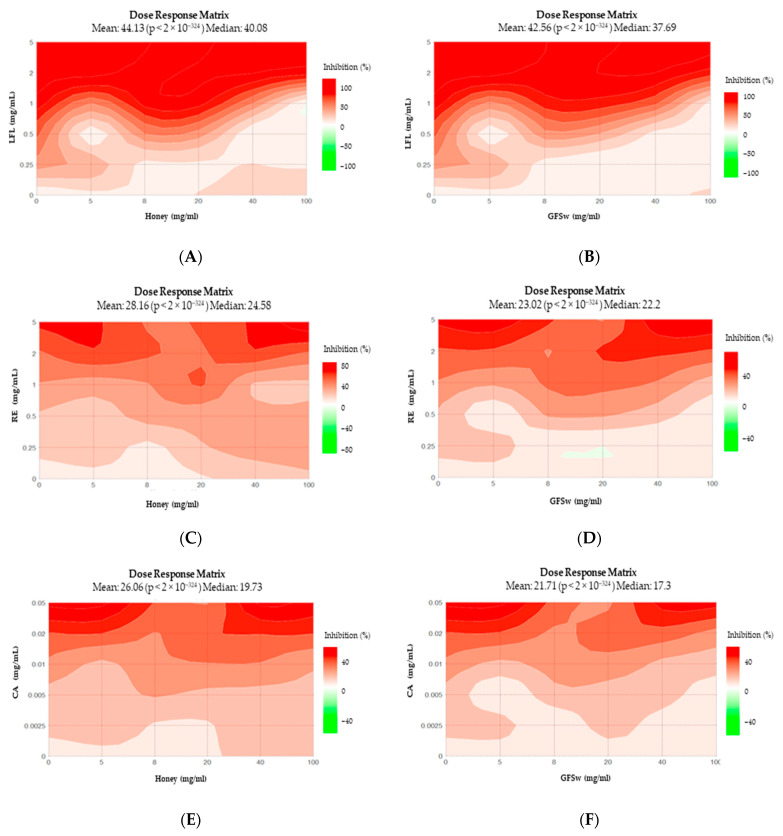
Dose–Response Matrix of DPPH, generated by SynergyFinder for (**A**) H_LFL; (**B**) GFSw_LFL; (**C**) H_RE; (**D**) GFSw_RE; (**E**) H_CA; (**F**) GFSw_CA; (**G**) H_EP; (**H**) GFSw_EP.

**Figure 6 antioxidants-12-01678-f006:**
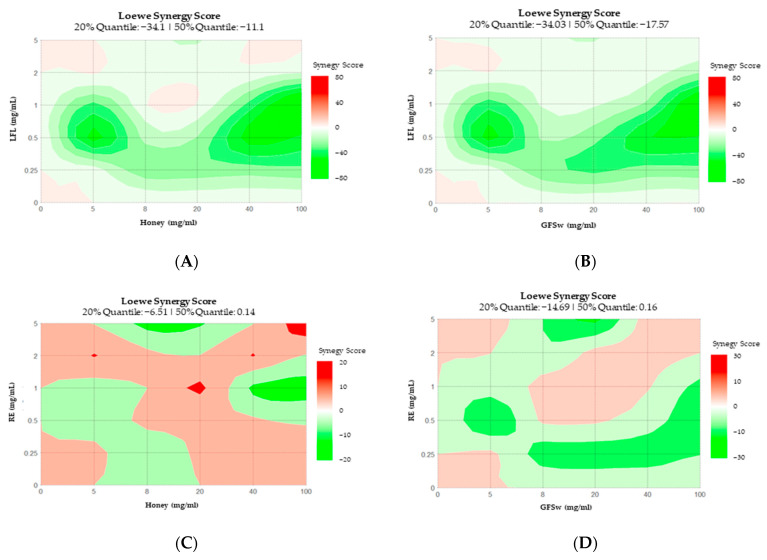
Loewe Synergy Score of DPPH generated by SynergyFinder for: (**A**) H_LFL; (**B**) GFSw_LFL; (**C**) H_RE; (**D**) GFSw_RE; (**E**) H_CA; (**F**) GFSw_CA; (**G**) H_EP; (**H**) GFSw_EP. Red represents synergism, and green represents antagonism.

**Figure 7 antioxidants-12-01678-f007:**
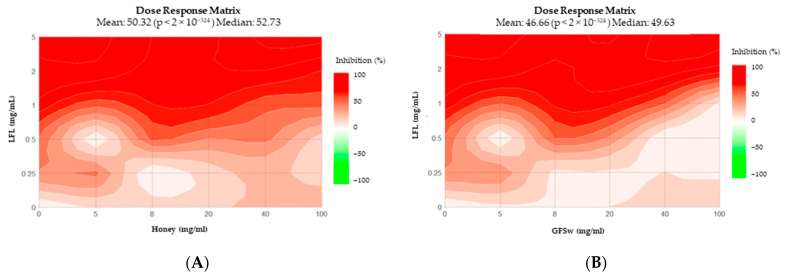
Dose–Response Matrix of ABTS, generated by SynergyFinder for (**A**) H_LFL; (**B**) GFSw_LFL; (**C**) H_RE; (**D**) GFSw_RE; (**E**) H_CA; (**F**) GFSw_CA; (**G**) H_EP; (**H**) GFSw_EP.

**Figure 8 antioxidants-12-01678-f008:**
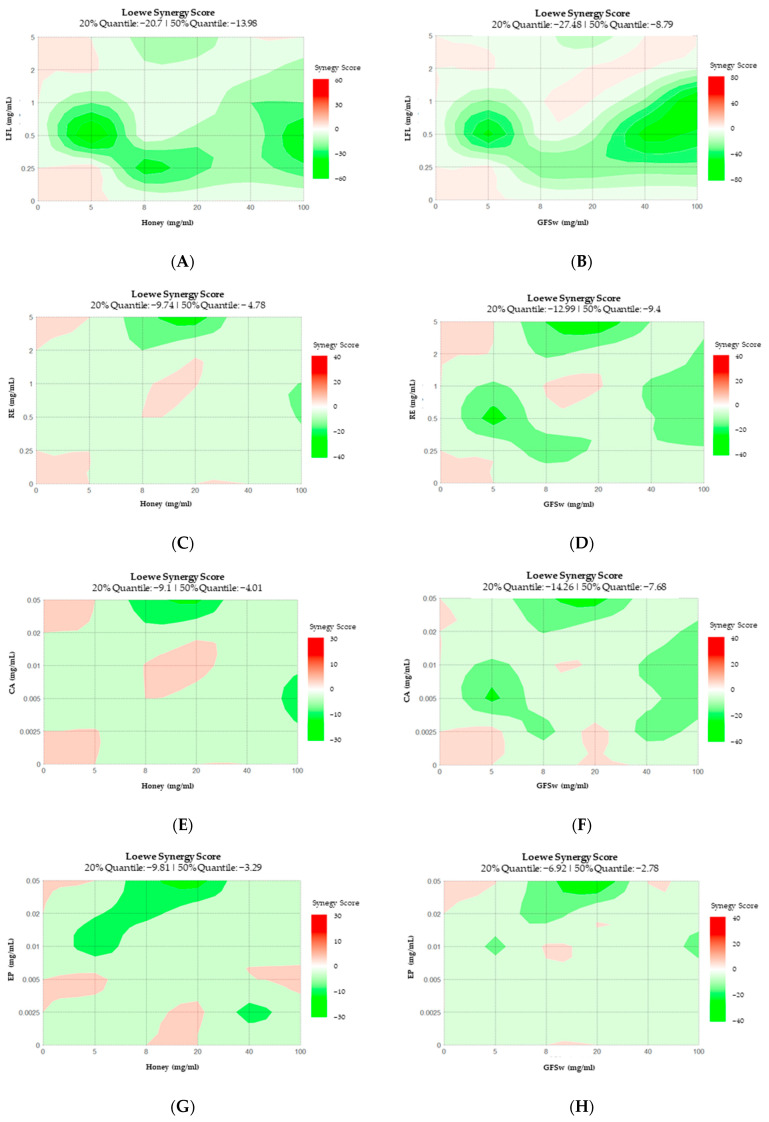
Loewe Synergy Score of ABTS generated by SynergyFinder for: (**A**) H_LFL; (**B**) GFSw_LFL; (**C**) H_RE; (**D**) GFSw_RE; (**E**) H_CA; (**F**) GFSw_CA; (**G**) H_EP; (**H**) GFSw_EP. Red represents synergism and green antagonism.

**Figure 9 antioxidants-12-01678-f009:**
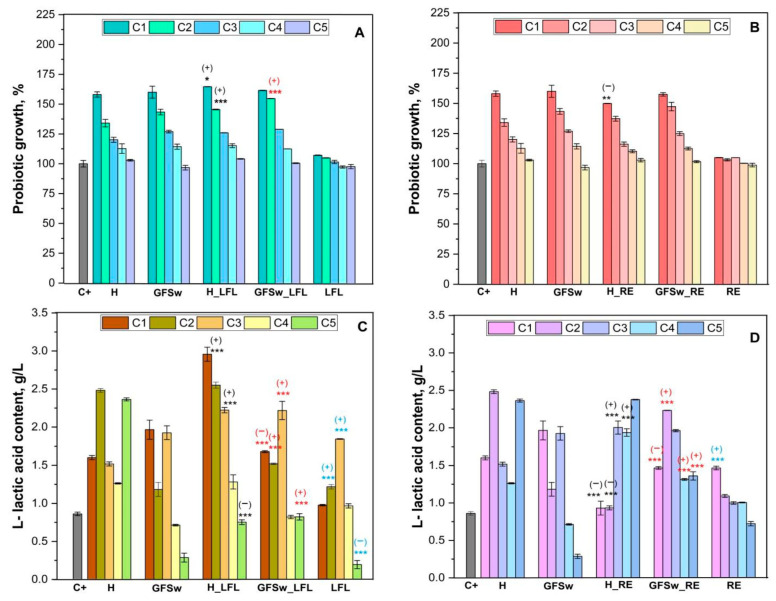
Probiotic growth-promoting (**A**,**B**) and L- lactic acid content (**C**,**D**) of honey (H), its biomimetic DES (GFSw), honey enhanced with honeysuckle flowers extract (H_LFL), and with raspberry extract (H_RE), GFSw enhanced with honeysuckle flowers extract (GFSw_LFL), and with raspeberry extract (GFSw_RE), honeysuckle flowers extract (LFL) and raspeberry extract (RE), C1—45 mg/mL; C2—25 mg/mL; C3—10 mg/mL, C4—5 mg/mL, C5—1 mg/mL for H, GFSw, H_LFL/H_RE, GFSw_LFL/GFSw_RE and C1 − C5—2.25–0.05 mg/mL for LFL and RE, ± error bars, α < 0.05, n = 3, *—σ between 0.05 and 0.01, **—σ between 0.01 and 0.001, ***—σ < 0.001; Black stars indicate statistically significant values oh H_LFL/H_RE compared to H, Red stars indicate statistically significant values of GFS compared to GFSw_LFL/GFSw_RE, blue stars indicate statistically significant values of LFL/RE compared to C+ (the strain of *L. reuteri* without any supplements); (+)—prebiotic activity; (−) inhibition.

**Figure 10 antioxidants-12-01678-f010:**
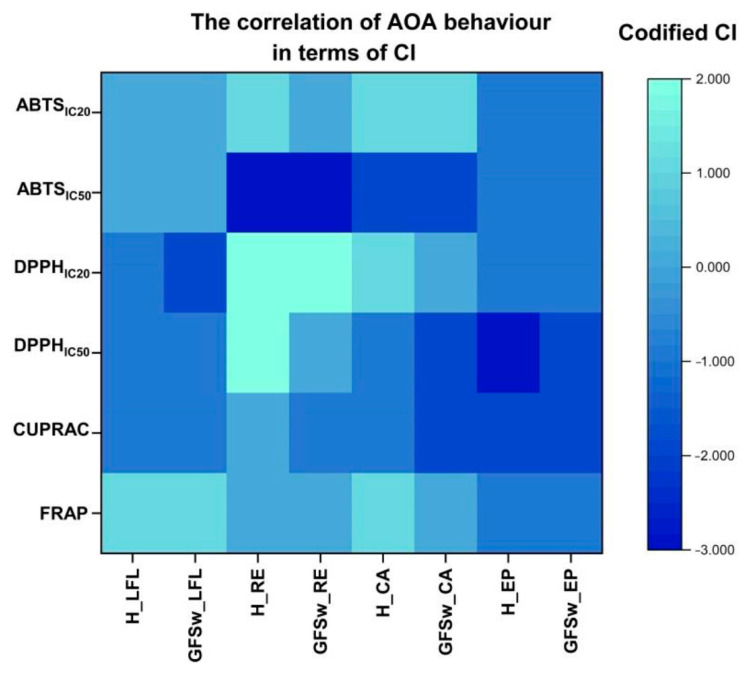
The Heatmap of the correlation of AOA in terms of CI.

**Table 1 antioxidants-12-01678-t001:** The results of TPC, TFC, HAT, and TAC of *L. caprifolium*.

Sample	TPC GAE mg/100 g DW	TFC QE mg/100 g DW	HAT ChaE mg/100 g DW	TAC, mg cya, 3-Glu Equivalent/100 g DW
*L. caprifolium* flower	651.79 ± 5.11	64.56 ± 2.12	587.38 ± 1.19	4.926 ± 0.011

GAE—gallic acid equivalent, QE—quercetin equivalent, ChaE—chlorogenic acid equivalent, 3-Gluequivalent—cyaniding 3-glucoside, DW—dried weight.

**Table 2 antioxidants-12-01678-t002:** The polyphenol compound from honeysuckle flowers by HPLC analysis.

Polyphenols	*Lonicera caprifolium* Flowers mg/g DW
Phenolic acids	
Caffeic acid	36.54 ± 0.04
Ferulic acid	1.72 ± 0.02
p-coumaric acid	0.46 ± 0.001
Chlorogenic acid	2.45 ± 0. 11
Flavonoids	
Epicatechin	2.83 ± 0.02
Apigenin	1.47 ± 0.007

**Table 3 antioxidants-12-01678-t003:** The combination index of honey (H_LFL) and its biomimetic DES (GFSw_LFL) mixtures with honeysuckle extract.

AOA Method	H_LFL	GFSw_LFL
FRAP	0.86 ±0.038	0.85 ± 0.029
CUPRAC	1.287 ± 0.020	1.195 ± 0.075
DPPH IC_50_	1.157 ± 0.036	1.346 ± 0.19
DPPH IC_20_	1.158 ± 0.038	1.536 ± 0.071
ABTS IC_50_	1.026 ± 0.015	1.093 ± 0.039
ABTS IC_20_	1.028 ± 0.014	1.093 ± 0.02

H_LFL—mixture of honey with honeysuckle flowers extract, GFSw_LFL—mixture of the NaDES GFSw—honeysuckle flowers extract. The terms IC_50_ and IC_20_ represent the analysis of concentration at 50% and 20% substrate inhibition, respectively.

**Table 4 antioxidants-12-01678-t004:** Quantitative data of the antioxidant activity as a function of RE and LFL concentrations.

Methods	RE	RE in H_RE Mixture	RE in GFSw_RE Mixture	LFL	LFL in H_LFL Mixture	LFL in GFSw_LFL Mixture
DPPH IC_50_ (mg/mL)	3.89 ± 0.12 ^c^	1.94 ± 0.13 ^b^	3.79 ± 0.21 ^c^	0.43 ± 0.011 ^a^	0.50 ± 0.02 ^a^	0.67 ± 0.15 ^a^
ABTS IC_50_ (mg/mL)	2.25 ± 0.11 ^b^	4.68 ± 0.24 ^c^	5.98 ± 0.95 ^d^	0.32 ± 0.01 ^a^	0.32 ± 0.01 ^a^	0.35 ± 0.02 ^a^
TEAC_FRAP_	94.72 ± 1.45 ^a^	121.81 ± 6.71 ^b^	93.12 ± 3.11 ^a^	263.81 ± 15.41 ^c^	293.41 ± 0.80 ^d^	296.58 ± 0.31 ^d^
TEAC_CUPRAC_	0.1 ± 0.004 ^a^	0.1 ± 0.013 ^a^	0.1 ± 0.009 ^a^	0.24 ± 0.012 ^b^	0.23 ± 0.039 ^b^	0.22 ± 0.0006 ^b^

Different letters show statistically different differences (±error bars, σ < 0.05, n = 3).

**Table 5 antioxidants-12-01678-t005:** Quantitative data of the antioxidant activity as a function of total mixture concentrations.

Methods	H_RE	GFSw_RE	H_LFL	GFSw_LFL
DPPH IC_50_ (mg/mL)	40.76 ± 2.89 ^b^	79.58 ± 4.45 ^c^	10.59 ± 0.48 ^a^	14.09 ± 1.18 ^a^
ABTS IC_50_ (mg/mL)	98.28 ± 2.96 ^b^	125.49 ± 11.56 ^c^	6.81 ± 0.16 ^a^	7.34 ± 0.22 ^a^
TEAC_FRAP_	6.07 ± 0.34 ^b^	4.63 ±0.16 ^a^	14.59 ± 0.026 ^c^	14.77 ± 0.025 ^c^
TEAC_CUPRAC_	0.0061 ± 0.0006 ^a^	0.0049 ± 0.0004 ^a^	0.0114 ± 0.0019 ^b^	0.0112 ± 0.00012 ^b^

Different letters show statistically different differences (±error bars, σ < 0.05, n = 3).

## Data Availability

All the data are contained within the article.

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
