# Peer review of "The Antioxidant and Prebiotic Activities of Mixtures Honey/Biomimetic NaDES and Polyphenols Show Differences between Honeysuckle and Raspberry Extracts"

_antioxidants, 2023, doi:10.3390/antiox12091678_

Round 1

Reviewer 1 Report

Manuscript ID: antioxidants-2534514

Title: The antioxidant and prebiotic activities of mixtures honey/ biomimetic NaDES and polyphenols show differences between honeysuckle and raspberry extracts

Although, the results are interesting, some details should be improved and explained. I would like to make some comments that authors could take into account to improve the overall quality of the manuscript.

Comments:

Abstract: The abbreviations as “NaDES” and “GFSW” should be explain. The authors use “GFSW” or “GFSw” in the manuscript, this abbreviation should be unified.

The “DES” and “GFSw” are used as synonyms, maybe it will be better to use in manuscript only one of them.

Line 42: The term „natural deep eutectic solvent (NaDES)” was introduced but unfortunately the advantages or disadvantages of this kind of food was not explain. Please, add a few sentences which explain influence of DES on nutritional value of food.

Lines 52-61: The appropriate literature should be given to support this statements.

Line 195: The extraction method of honeysuckle flowers should be described shortly.

Line 367: It will be enough if the second decimal place of the retention time will be shown.

Line 390: Table 2 or 3?

Figure 1 and 2 are not described clearly. These Figures should be explain step by step, some explanations should appear earlier, for example explanation “In accordance with our previous study [4], we have categorized the CI values for ease of comparison as follows: 0.5-0.7 indicates strong synergism, 0.7-0.9 denotes moderate synergism, 0.9-1.1 implies nearly additive behaviour, 1.1-1.5 signifies moderate antagonism, 1.5-2 indicates moderate to strong antagonism, and CI >2 represents strong antagonism.”

This explanation “However, if the theoretical curve lies below the experimental curve, it suggests synergism, whereas if the theoretical curve is higher than the experimental curve, it indicates antagonism.” should be introduce earlier.

Line 472: It was written that “the majority of samples exhibited a positive impact on bacterial growth compared to the control”. Control = C+? I think that only samples with the same concentration of sugars can be compared. It means that C+ with LFL samples and RE samples. I see that H is a control for H_LFL, etc. But it is not clear in mentioned sentence. The composition of “C+” should be defined in methodological section.

Line 486 -489: The higher probiotic growth was not proved, the differences were to small (lines 478-479: “The growth percentages were 145.383 ± 0.29% for H_LFL compared to 134.076 ± 3.099% for simple honey, and 154.688 ± 0.13% for GFSw_LFL compared to 143.382 ± 2.36% for GFSw.” I am very surprised that standard deviation was so small (RSD = 0.13/154.68*100% = 0.08%).

Have the data presented in Table 4 been presented in other manuscript (samples with RE extract)?

The meaning of number presented in Table 5 should be described clearly.

Reviewer 2 Report

This manuscript presents an investigation of differences between  honeysuckle and raspberry extracts based on antioxidant and prebiotic activities of mixtures honey/ biomimetic NaDES and polyphenols.

What definitely needs to be changed is the extremely large similarity in the form of presentation of the results as in your previous work (ref [4]), and then there is the following:

L342 +/- sign is correct (bease on the statement on L340

Figure 1B - place the legend also in the left corner as on other figures. In figure 1 B the secindar y-axis is not readable

Table 4 - why are listed only 2 AO methods (from four that were conducted)

The presentation of the results in Table 5 is extremely confusing (eg it is not clear why some fields are white while others have a grey-blue background). If you want to use colors in the presentation of the results - I suggest heatmaps.

Sincerely

Reviewer 3 Report

Ref. The antioxidant and prebiotic activities of mixtures honey/ bio-mimetic NaDES and polyphenols show differences between honeysuckle and raspberry extracts.

The authors measured divers characteristic of honey / NaDES - honeysuckle (Lonicera caprifolium, LFL) extract and compared it with the mixture honey / NaDES -raspberry extract.

Modern methods have been employed to measure, eg. total polyphenol content, total hydroxycinnamic acid content, total flavonoid content, anthocyanin content, radical scavenging activity, etc.

The methods are sufficiently described.

Specific comments

Key words. You should not to repeat the words from the article title. Therefore, remove antioxidant activity . Add – Lonicera caprifolium

The aim of study is not clearly described. Please, try to be more specific. May be the hypothesis can be present.

In several places (indicated below) the results description should be corrected.

L 36 – citation , please.

L 64-65 – It is obvious that Lonicera caprifolium belongs to the genus Lonicera. Please, remove the information. It is trivial. Lonicera caprifolium belongs to the family Caprifoliacea. It is a point.

L54-61. Please, add the literature citations. It is a long statement and should be  should be accompanied by quotes.

L 92 _ give full name of the plant. Moreover, add the botanical family name for the Lonicera species.

L.92- Can you add some more characteristic to the multifloral honey (i.e. types of species that were used to produce the honey)

L 350-354 – This statement should be removed from Results section to the Discussion section. You should not compare your own data with literature. Just describe your results.

L 355- 361 – as above. Do not compare your data with literature in the Results section.

L 362 Table 1- add the Latin name of the flower

L 370- 375 - Do not compare your data with literature in the Results section.  

L 503-509 – remove to the Discussion section. Here, no results have been described.

L 529  should be …. “from our previous results”,  not previously  

L 542 Table 4 – I can not understand  why this table is described in Discussion section, it should be inserted and described in Result section.

Reviewer 4 Report

The article deals with the fortification of mixtures honey/ bio-2 mimetic NaDES with honeysuckle and raspberry extracts, considering few different dimensions, such as polyphels and flavonoids content, antioxidant activity, prebiotic and probiotic activities. Quite surprisingly, honeysuckle was generally more effective than raspberry,  which was previously unknown (at least to my knowledge), opening interesting scientific and practical perspectives.

The study is very well organized, as well as the experiments, and the presentation of the results is fine.

There is little to add, except:

Comment 1: A very minor remark about the use of comma (,) instead of point (.) in line 122 (mixing 0,01 mL of honeysuckle).

Comment 2: In line 142 (Table 4): Indicating statistical significance directly inTable 4, for example with usual "*", would be useful.

That's all. Minor revision is appropriate.

English language is fine.

Round 2

Reviewer 1 Report

The paper has been corrected significantly and I think that final version of this paper can be considered for publication.

Reviewer 2 Report

All my comments have been taken into account, so now the manuscript is in a form that is acceptable for publication.

Sincerely